# A Novel Signal Restoration Method of Noisy Photoplethysmograms for Uninterrupted Health Monitoring

**DOI:** 10.3390/s24010141

**Published:** 2023-12-26

**Authors:** Aikaterini Vraka, Roberto Zangróniz, Aurelio Quesada, Fernando Hornero, Raúl Alcaraz, José J. Rieta

**Affiliations:** 1Biosignals and Minimally Invasive Technologies (BioMIT.org), Electronic Engineering Department, Universitat Politecnica de Valencia, 46022 Valencia, Spain; aivra@upv.es; 2Research Group in Electronic, Biomedical and Telecommunication Engineering, University of Castilla-La Mancha, 16071 Cuenca, Spain; roberto.zangroniz@uclm.es (R.Z.); raul.alcaraz@uclm.es (R.A.); 3Arrhythmia Unit, Cardiology Department, General University Hospital Consortium of Valencia, 46014 Valencia, Spain; quesada_aur@gva.es; 4Cardiovascular Surgery Department, Hospital Clínico Universitario de Valencia, 46010 Valencia, Spain; hornero_fer@gva.es

**Keywords:** photoplethysmography, motion artifacts, noise detection, signal reconstruction, health-tracking, uninterrupted monitoring, heart-rate, heart-rate variability, pulse rate variability

## Abstract

Health-tracking from photoplethysmography (PPG) signals is significantly hindered by motion artifacts (MAs). Although many algorithms exist to detect MAs, the corrupted signal often remains unexploited. This work introduces a novel method able to reconstruct noisy PPGs and facilitate uninterrupted health monitoring. The algorithm starts with spectral-based MA detection, followed by signal reconstruction by using the morphological and heart-rate variability information from the clean segments adjacent to noise. The algorithm was tested on (a) 30 noisy PPGs of a maximum 20 s noise duration and (b) 28 originally clean PPGs, after noise addition (2–120 s) (1) with and (2) without cancellation of the corresponding clean segment. Sampling frequency was 250 Hz after resampling. Noise detection was evaluated by means of accuracy, sensitivity, and specificity. For the evaluation of signal reconstruction, the heart-rate (HR) was compared via Pearson correlation (PC) and absolute error (a) between ECGs and reconstructed PPGs and (b) between original and reconstructed PPGs. Bland-Altman (BA) analysis for the differences in HR estimation on original and reconstructed segments of (b) was also performed. Noise detection accuracy was 90.91% for (a) and 99.38–100% for (b). For the PPG reconstruction, HR showed 99.31% correlation in (a) and >90% for all noise lengths in (b). Mean absolute error was 1.59 bpm for (a) and 1.26–1.82 bpm for (b). BA analysis indicated that, in most cases, 90% or more of the recordings fall within the confidence interval, regardless of the noise length. Optimal performance is achieved even for signals of noise up to 2 min, allowing for the utilization and further analysis of recordings that would otherwise be discarded. Thereby, the algorithm can be implemented in monitoring devices, assisting in uninterrupted health-tracking.

## 1. Introduction

Photoplethysmography (PPG) is a versatile technology, with applications ranging from user authentication and remote device control to health monitoring [1,2,3]. Compared to electrocardiograms (ECGs), PPG hardware is simpler, more cost-effective, and less power consuming [1,4]. PPG devices are compact and non-invasive, while one of their biggest advantages is allowing freedom of movement during data collection, hence being ideal for remote monitoring [1,2,5,6]. Key health parameters frequently monitored with PPG include heart-rate (HR), HR variability (HRV), oxyhemoglobin saturation, blood pressure, and respiratory rate analysis [4,5,7,8].

PPG is an optical technique that consists of a light source and a photodetector with a simple operation: the source emits light to the tissue, and the photodetector converts the reflected or transmitted light from the tissue to the hardware into an output current, forming the PPG signal [1,2,4,8]. The light source can be either an infrared (IR)/near-IR light-emitting diode (LED) or a green LED [4]. Even though an IR LED is more effective in penetrating the skin tissue, over the last years, a green LED is preferred due to its lower susceptibility to artifacts [4,9].

The amount of light received by the photodetector is strongly affected by tissue vascular characteristics and is inversely proportional to blood volume, which varies according to the cardiac cycle, creating the AC-like waveform of the PPG signal [1]. PPG also contains a quasi-DC component, influenced by respiration, sympathetic nervous system activity, and thermoregulation [1,4,9,10]. Both components can be further analyzed after appropriate filtering in order to extract health-related indices [1].

In addition to contact-based PPG, remote PPG has emerged as an alternative modality for extracting health-related information. Remote PPG relies on contactless imaging devices that capture subtle color changes in the skin to derive physiological parameters. This approach gained significant traction, especially during the COVID-19 era, due to concerns about potential infections transmitted through skin contact [11]. Subsequently, remote PPG has been widely adopted, with numerous methods and applications offering an easily accessible and non-contact assessment of health metrics, including heart rate and SpO${}_{2}$ estimations [11].

Factors affecting PPG signal quality include changes in tissue properties due to muscle movement, sensor displacement, loose band attachment, and environmental factors such as ambient light [4,5,12]. Movement is the most common source of artifacts, called motion artifacts (MAs) [1]. To mitigate these issues, typical PPG denoising methods involve band-pass filtering (0.5 to 15 Hz) to preserve heart rate frequency (0.83 to 3.33 Hz) while removing baseline fluctuations due to respiration (0.13 to 0.67 Hz) [13]. Notwithstanding, wavelet decomposition or comb filtering might be more efficient preprocessing methods leading to a higher signal-to-noise ratio (SNR) [14,15]. It is worth noting that the frequency spectrum of MAs can overlap with HR frequencies, necessitating additional processing for their detection and removal, which remains a challenging task [4,5,7,12,13,16].

A vast number of studies has dealt with artifact detection in PPGs [12,17,18]. Algorithms for artifact detection and/or removal can be classified into three main categories [12]. The first category includes the signal decomposition methods, based on the suppression of noisy components at different frequency bands [12,17,18]. Common signal decomposition methods are discrete wavelet transform, empirical mode decomposition, independent component analysis, variational mode decomposition, and short-time Fourier transform [12]. Although signal decomposition methods per se do not need any references to operate, in practice they utilize additional signals such as ECGs or different-wavelength PPGs in order to detect MAs in particular and define the usability of the corrupted signal [17,18].

The second category consists of adaptive filtering methods such as the Wiener filter, least mean squares, and recursive least squares [12,16,19]. These methods require additional signals to be used as a reference for the noisy components to be removed [12]. Despite their popularity regarding MA detection and removal, evidence shows that Accelerometer (Acc) and Gyroscope (Gyro) signals are not always reliable motion references [20]. Moreover, these types of signals do not allow for the detection of MAs due to negligible movements such as finger tapping.

The final category is actually a medley of variable methods other than the ones described above [12]. This includes classic spectral analysis, when the frequency spectrum of the noisy source does not overlap with the HR frequency, an analysis of statistical or morphological parameters of PPG signals, or machine learning techniques [7,12,21,22,23]. Besides the unsuitability of some of those methods when MAs are the main source of noise, they often require additional signals as references [21,22,23]. Methods and strategies borrowed from other fields can also be useful in MA detection. Such an example is the self similarity matrix analysis, which allows for the detection of abrupt morphological changes, by performing signal analysis in the feature space [24]. Although not directly applied in PPG recordings, given the periodic nature of the signals and a sufficiently short segment selection, this method could work exceptionally in detecting MAs. The problem of signal usability, however, would still remain.

In addition to the aforementioned limitations regarding the existing algorithms to detect MAs or other kinds of PPG noise, noisy signals are often discarded or partially used after signal quality analysis [17,22]. Nevertheless, continuous health monitoring requires the constant supply of clean data to be further processed in order to extract health-related indices. Although some studies have attempted uninterrupted HR monitoring with low-intensity MA PPGs, deviant results and the further use of additional signal sources necessitate less-dependent and more efficient techniques [12,16,19].

The present study introduces a method that is able to detect PPG artifacts without the need for auxiliary signals, further reconstructing PPGs by utilizing information from adjacent clean segments. This approach enables the continuous monitoring of HR, even in the presence of MAs. The method is tested under clinical PPG signals containing MAs as well as synthetic signals with varying levels of added noise. The results suggest that its implementation in monitoring devices can provide usable data for seamless continuous health tracking. The remainder of the document is as follows. Section 2 presents the database utilized as well as the preprocessing methods and explains the algorithm and the methods for verification of the presented techniques. Section 3 presents the results, while Section 4 goes further into the findings of the study. Finally, the general conclusions derived are reported in Section 5.

## 2. Materials and Methods

### 2.1. Materials and Preprocessing

For the validation of the presented method, the BIDMC PPG and Respiration Dataset [25] was used. This dataset contains eight-minute recordings from 53 critically ill patients extracted from the MIMIC II matched waveform Database [26,27]. Each recording contains physiological signals including PPGs and ECGs. Recordings were exported with a sampling rate of 125 Hz and were resampled to 250 Hz in order to increase resolution. All analysis steps were performed in a Matlab^®^ (2022) environment (Version R2022a, The MathWorks Inc., Natick, MA, USA).

Two distinct databases were then created for the validation of the noise detection and PPG reconstruction method.

1.**ECG match database (ECGMDB):** This database contains 30 clinical recordings from 16 patients within the BIDMC PPG and Respiration Dataset. Recordings consisted of a noisy PPG segment with a maximum duration of 20 s, alongside a corresponding clean ECG segment. The aim of this database was to evaluate whether HR analysis from reconstructed PPGs matched the HR analysis from the ECGs, determining the reliability of such reconstructed PPGs for continuous HR monitoring, especially in those cases where the ECG is unavailable.2.**PPG polluted database (PPGPDB):** For this database, 28 real 8-min clean PPG recordings of 17 patients from the BIDMC PPG and Respiration Dataset were recruited. For each real PPG, 22 synthetic signals were created. Eleven of these signals were created from the addition of random noise with a varying duration of 2, 5, 10, 20, 30, 45, 60, 75, 90, 105, and 120 s in the real PPG and the remaining 11, with the addition of random noise with varying duration of the same length, following the complete cancellation of the respective clean PPG segment. A scheme of the utilized databases can be observed in Figure 1. The objective of the creation of two synthetic sub-databases out of the PPGPDB was to test the noise detection algorithm both in the case of deteriorated signal quality due to noise and in the case when the signal is completely lost due to loss of contact with the recording device.

For the synthetic signals with and without signal cancellation, the amplitude of noise was set equal to the amplitude of the PPG signal. Nevertheless, various amplitudes of noise have been tested without a difference in MA detection accuracy, as can be observed in Figure 2. For the replacement of the clean PPG segment by random noise, first, the root mean square (RMS) of the signal was calculated. White Gaussian noise of the mean SNR of −3 dB was then added to a segment of equal values of RMS/2 of the signal, leading to signals with morphology as seen in Figure 3b. For the addition of random noise to the originally clean signal, white Gaussian noise of the mean SNR of 12.89 dB was added to the clean signal segment, as can be seen in Figure 3c.

Preprocessing of the ECG signal included powerline interference, muscle noise, baseline wander removal and R-peak detection, as described elsewhere [28,29,30]. PPG preprocessing started with a 2nd-order high-pass Butterworth filter with a 0.5 Hz cut-off frequency to remove the baseline fluctuation due to respiration. Afterwards, high frequency noise was removed using a three-level discrete wavelet transform based on Coiflets, a quadratic loss function, and fine-scale noise variance estimation, which was found to preserve the PPG signal morphology according to visual inspection.

After signal preprocessing, the main analysis was performed. The main analysis consists of two phases: the motion artifact detection phase and the signal reconstruction phase, each described in detail.

### 2.2. Motion Artifact Detection

The first phase of signal reconstruction is the MA detection. The developed and presented algorithm makes exclusive use of PPG signals in order to detect MAs or other types of noise. A breakdown of the followed steps for this part of the algorithm is presented.

**Step 1:** The first step of MA detection was signal segmentation into 8 s epochs with 75% overlapping. This allows for the detection of noise at a minimum length of 2 s.**Step 2:** Spectral analysis was performed in order to detect the dominant frequency (DF) of each epoch [31]. Given that normal HR rarely exceeds the 0.83–3.33 Hz limits, a first-level detection of artifacts was performed by verifying that DF belongs to the broadened 0.3–4 Hz range. Once this hypothesis was verified, the 2nd ($h_{2}$) and 3 ($h_{3}$) harmonics were defined as 2 × DF and 3 × DF, respectively, and the spectral power (${\mathrm{SP}}_{i}$) of each 0.7 Hz band with DF, $h_{2}$, and $h_{3}$ centered was calculated.**Step 3:** The cumulative spectral power (CSP) was defined as the sum of the spectral power at each of the three spectral windows
(1)$$\mathrm{CSP}={\mathrm{SP}}_{1}+{\mathrm{SP}}_{2}+{\mathrm{SP}}_{3}$$
and compared with the total spectral power (TSP) of the segment. An artifact was detected when the condition CSP<0.65×TSP was fulfilled.**Step 4:** In the case in which no artifact was detected, a last-round check was performed by verifying the existence of a spectral peak at each of the 3 spectral windows of DF, $h_{2}$, and $h_{3}$. In the opposite case, the segment was labelled as noise. The aim of this step is to detect any artifacts with a spectral peak close to the DF of a normal heart rate and a gradual spectral slope. Figure 4c illustrates the power spectral density of a clean segment, while Figure 4d shows the power spectral density of a noisy segment.
Figure 4Power spectrum of a clean (**a**) and a noisy photoplethysmography (PPG) signal segment (**b**) and their corresponding PPG signals. Shaded areas indicate the 0.7 Hz spectral windows around dominant frequency (DF) and the harmonics. Even though cumulative spectral power (CSP) is relatively high in case (**d**), no peak is found in the window with h2 centered. Spectral overlapping between two spectral windows is indicative of noisy segments.
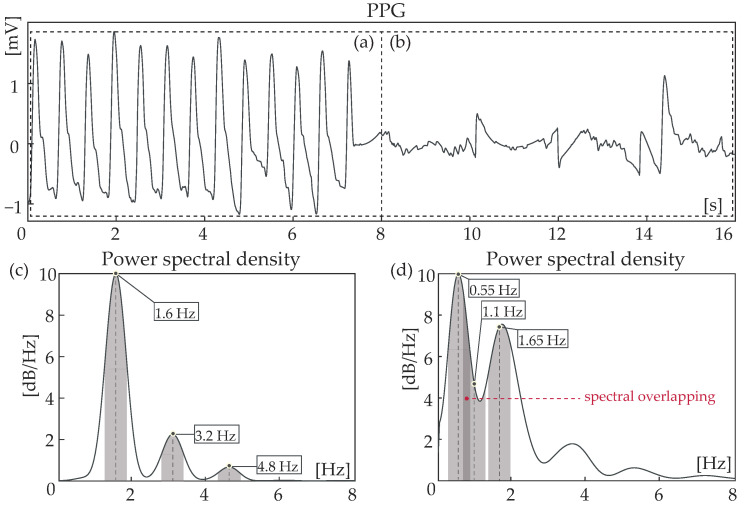

**Step 5:** Each 8 s segment detected as noise was then stored into a matrix, and the actual noisy part of each segment was finally defined by checking whether sequential windows are detected or not as noise. Figure 5 shows the power spectral density of a segment that contains a clean and a noisy part. Originally, the entire 8 s segment is labelled as noise, but in the end, only the first four seconds are detected as actual noise, and the remaining four seconds are considered as a clean PPG signal.
**Step 6:** A final optional amplitude control was performed in order to spot undetected noise of very low or very high amplitude. For the very low and the very high amplitude noise, the following thresholds were used, respectively, after trial and error
(2)$$Th_{min}=\frac{\mathrm{RMS}(nPPG)}{1.2},$$
where nPPG is the normalized PPG according to the highest amplitude and
(3)$$Th_{max}=3.3\times \mathrm{RMS}(PPG).$$

Once the noise detection is finished, the algorithm extends the noise labeling by an amount of sample points proportional to the sample frequency at the first and the last part of each noisy segment, in order to deal with undetected noise in the beginning and the end of a segment due to the 2 s noise detection resolution. Practically this means that the noisy part is extended by one second (normally 0.5–2 pulses, depending on HR) at the beginning and the end of each noisy segment. Finally, in cases in which a clean segment less than 2× the sampling frequency is detected between two noisy parts, the clean segment is also labelled as noise. Figure 6 shows the block diagram of the main MA detection algorithm.
Figure 5An 8 s epoch containing a noisy (the first 3.5 s) and a clean (the remaining 4.5 s) part. The entire segment was originally detected as noise (no peak in the spectral window of the 2nd harmonic), but after sequential analysis with the adjacent segments, only the first 4 s were labelled as noise. PPG: photoplethysmography.
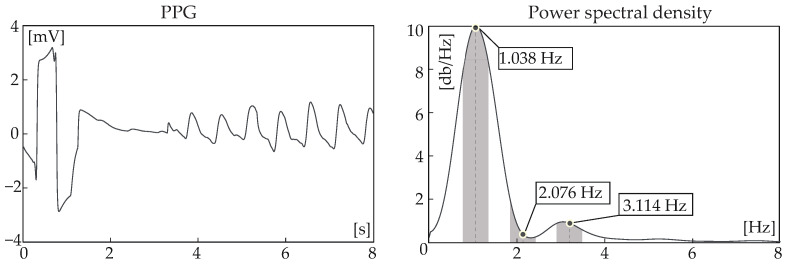



### 2.3. Signal Reconstruction

The signal labelled as noise was fed to the signal reconstruction algorithm. A block diagram illustrating the fundamental steps of the noise reconstruction algorithm can be seen in Figure 7. The entire signal reconstruction analysis was based on the information acquired from the clean signal segments close to noise. Each step is explained in detail.

**Step 1—Calculating the reference parameters:** Before reconstruction, a short preparation including the calculation of parameters that will serve as a reference for the signal reconstruction were computed. These include the following:-The HR of the clean segments surrounding noise. PPG pulse peaks were detected by 2nd-derivative analysis based on an adaptive amplitude threshold [32]. Each segment consisted of 10 pulses.-The noisy signal was divided into two segments of equal length, and each segment was band-pass filtered with a 2nd-order Butterworth filter with cut-off frequencies ±0.35 of the DF according to the HR of the closest detected pulse.-The DF of each segment was calculated and used as the baseline for the signal reconstruction (baseline DF). In order to avoid any distortion in the morphology of the signal between the clean-reconstructed signal transition, the first and last points of the noisy segments were appropriately relocated so that they correspond to the middle point of the valley between two successive zero-crossing points.-HRV and amplitude variability of the closest clean segments were calculated as the difference in HR and amplitude between two successive peaks, respectively.-The baseline pulse for each segment was calculated as the median pulse of the 5 closest pulses of the clean signal surrounding the noise. In order for the signal to be coherent, each baseline pulse started and ended in the middle point of the valley between two successive zero-crossing points.**Step 2—Localizing the peaks of the segment under reconstruction:** The reconstruction process was performed in segments of 10 s or less, starting from the edges and moving towards the center of the noisy part. For the HR calculation of each peak, the baseline DF of each segment along with the corresponding HRV was considered, so that the HRV of the closest peak was assigned to the first pulse, the HRV of the second closest peak was assigned to the second pulse, and this process continued until the HR of all peaks of this segment was calculated. For the peak locations, the HR distance from the last and first detected peaks of the clean segments surrounding the noise was calculated, and the remaining peak locations were defined from the distance to the last detected peak according to HR.**Step 3—Localizing the peaks of the remaining signal:** This step was performed only if the noise duration was longer than 20 s. The signal left to be reconstructed was calculated by considering PPG HR and the location of the last detected peaks of the already reconstructed segment. The next segment under reconstruction was then defined and an iterative peak location process started using the baseline DF as well the HRV of the closest pulses of the already reconstructed segment.**Step 4—Fixing inconsistencies in peak localization:** When no 10 s segment was left for reconstruction, a last-round control for the signal left for reconstruction was performed by calculating the distance between the two middle peaks of the reconstructed signal. If the distance was longer than 0 but shorter than the length of a pulse, each distance between two successive peaks was prolonged by one sample point, starting from the middle of the reconstructed segment and moving simultaneously forward and backwards, until no sample points were left. If the distance was longer than a pulse, the corresponding number of pulses was added in the middle of the reconstructed segment. This step was repeated until the distance of the two middle peaks was equal to the length of a pulse, with a margin of 2 sample points.**Step 5—Signal replacement:** The reconstruction was finalized by replacing the noisy signal with the baseline pulses, prolonged or shortened according to the peak-to-peak distance, so that each peak location corresponded to the peak of each pulse. Additionally, each pulse was stretched or shrunk in amplitude according to the corresponding amplitude variability.

Figure 8 shows an example of a clean signal, the signal after noise pollution (both ways), and the same signal after signal reconstruction. The reconstructed signal is shown twice. Once in a continuous red line and once in a dashed red line (overlapping with the original PPG) to facilitate a visual comparison with the original PPG. A high similarity between the morphology of the original and reconstructed signals is shown. Likewise, Figure 9 shows the signal reconstruction outcome of a real, noisy signal and its corresponding ECG.

### 2.4. Evaluation

The evaluation of the proposed methodology consists of two stages: the evaluation of the noise detection method and the evaluation of the signal reconstruction method. The evaluation of the noise detection method is similar for both databases, where a comparison between manual annotations and annotations of the algorithm was performed. For the PPG reconstruction method, the comparison between original and reconstructed signals was performed by comparing the calculated HR, with the reference varying according to the database. For the ECGMDB, a less straightforward reference was used, as no clean PPGs were available. For this reason, the HR of the ECGs was utilized, including some sort of variation due to natural variance between ECG and PPG signals. The evaluation of the PPGPDB set the HR of the original PPGs as the baseline, this modality consisting of a more clear and direct comparison.

#### 2.4.1. Evaluation of the Noise Detection Method

For both databases, the first step of the evaluation was the manual annotation of the noise by the visual inspection performed by two experts. Given the fact that the PPG waveform is clear and simple, there was consensus regarding the noise starting and end points in all recordings. In any case, a margin of 2 s before and after each noisy segment was allowed, to account for any minor discrepancies of 1 ms or less during noise annotation as well as the corrections and extension of noise labeling performed by the algorithm. Confusion matrices were created for each recording, where each point per second of the signal was assigned to one of the following categories:TP: 1 s of noise, correctly detected as noise;FP: 1 s of clean signal, wrongly labelled as noise;FN: 1 s of noise, wrongly labelled as clean signal;TN: 1 s of clean signal, correctly identified as clean signal.

In order to put more emphasis on the correctly identified or erroneously missed noise, TP and FN were weighted by a factor of ×50,×20,×10, and ×5 per second for the cases of 2, 5, 10, and 20 s of noise, respectively.

For both databases, normality and homoscedasticity were tested using Kolmogorov–Smirnov and Levene tests, respectively, indicating non-normal and non-homoscedastic results.

#### 2.4.2. Evaluation of the PPG Reconstruction Method

The efficiency of PPG reconstruction for ECGMDB was analyzed using the HR of the ECGs as the ground truth, while that for the PPGPDB was analyzed by using the HR of the original PPGs as the ground truth. The HR of the ECGs was determined by R-peak detection [30]. The HR of the PPGs was determined by pulse peak detection, using an adaptive amplitude second derivative method and a second-round control to delete any annotations set less than 0.35 ms apart [32]. A comparison between the HR in the ECGs and the reconstructed PPGs was performed with a Mann–Whitney U-test (MWU). Additionally, the HRs of each ECG and the reconstructed PPG signals were correlated with a Pearson correlation coefficient. The mean absolute errors in the HR calculation were also defined for each recording.

Finally, a Bland–Altman (BA) analysis was performed [33]. BA analysis is a method to assess comparability between two methods by studying the mean differences between them [33,34]. It plots the mean differences with respect to the average value of both methods. Nevertheless, if one of those methods is more important or trustworthy than the other, the relative differences using one of the two methods as a reference can be calculated. As the HR measured from ECGs is the ground truth in this analysis, it was used as a reference. As BA analysis can be interpreted according to the nature of the study, we quantified the results as the percentage (%) of recordings that fall within the confidence interval (CI), which is defined according to the mean and SD of the differences in calculation [34]. We set a threshold of α=0.9 (90%) to define an adequately measured feature from the reconstructed PPG, with the rationale that, if the HR from the ECGs and the PPGs is comparable for 90% of the recordings, the reconstructed PPG would be reliable in calculating the HR.

## 3. Results

### 3.1. MA Detection

The performance of MA detection can be seen in Table 1 and Table 2. For the ECGMDB and the subdatabase of PPGPDB including signal replacement by noise, results are shown in Table 1, as results were the same for all lengths of noise. For the subdatabase of PPGPDB including synthetic signals with noise addition, different results were obtained for each noise length, as can be seen in Table 2.

As ECGMDB included noisy recordings with lower signal quality, MA detection showed lower detection accuracy. In any case, however, detection accuracy was above 90% and a relative balance between false positive and false negative rate was observed, as can be deduced from the balanced sensitivity–specificity rates, with a shift towards false positive detection. On the other hand, detection performance was 100% for synthetic PPGs, including signal totally replaced by noise, regardless of the noise length. The difference between the performance of the two databases can be attributed to the higher signal quality of PPGPDB, significantly facilitating the noise detection with respect to the ECGMDB, where corrupted PPG parts of some ms could be observed across the recordings. Additionally, when the signal is completely cancelled and replaced by noise, the analysis is significantly facilitated, given the fact that the noise detection method is based on spectral analysis.

This observation is corroborated by the slightly attenuated yet almost perfect performance results of the PPGPDB subdatabase, where the signal was kept intact and noise was added. In this case, as the spectral content of the clean signal was preserved, the analysis showed a tendency to overdetect noisy segments, as can be seen from the 100% sensitivity rates in almost all cases and the slightly lower (less than 1%) specificity rates. In other words, the algorithm was able to detect the entire noisy segment in all circumstances, while in some cases one or more seconds of a clean segment surrounding noise were also labelled as noise, mostly due to the structure of the technique, slightly extending noise labeling and specifying a specific start and end point to later facilitate signal reconstruction. Notwithstanding, the accuracy results were higher than 99%, suggesting an excellent functioning of the technique.

### 3.2. PPG Reconstruction

#### 3.2.1. ECG Match Database

Table 3 shows the mean and median HR for the reconstructed PPG and the corresponding ECG segment as well as the corresponding absolute errors for the ECGMDB. It can be observed that the HR calculation is similar for both signals, as is also demonstrated from the HR patterns shown in Figure 10, which are almost identical for most of the recordings. The absolute error is close to 1 bpm (mean: 1.59 bpm; median: 0.67 bpm). The same table illustrates the correlation in HR calculation between ECGs and PPGs, which indicates an almost perfect correlation of 99.31%. Note that HR calculation is also analyzed from different sources (ECGs vs. PPGs). The MWU result does not indicate statistically significant differences in the HR calculation (p=0.8480). BA analysis shows that the HR calculation in 93.33% of reconstructed PPGs fell within the confidence interval, using the HR calculated from the ECGs as the ground truth. The same analysis, see Table 3, shows a coefficient of variation of 0.03%.

#### 3.2.2. PPG Polluted Database

Taking into consideration the fact that the same segment was chosen and processed both for the synthetic subdatabase including signal replacement by noise and for the synthetic subdatabase including noise addition while preserving the signal intact, the PPG reconstruction process was the same and led to the same results. Therefore, the results are presented once, including both subdatabases of the PPGPDB.

Figure 11 shows the box and whisker plots for the mean and SD of the HR calculated from the original signal (dark red box and whisker plot) and from the reconstructed signals for various noise lengths (2–120 s). The median values can be seen in Table 4. None of the results showed statistically significant differences with respect to the values of the original PPG (p>>0.05) according to the MWU test. Overall, the values between the original and reconstructed segments for all lengths of the reconstructed signal are very close. Specifically, when the HR of only the reconstructed segment is calculated (HR-noise or $HR_{n}$), the HR in the reconstructed segments is close to the HR in the original segment, with very small variations (difference: 1.19 bpm in the worst-case scenario). This is true even for the 120 s segments. Respectively, the HR calculation over the entire eight-minute recordings indicates almost identical values between the original PPG and the PPGs including reconstructed segments of different durations (maximum difference: −0.26 bpm).

The same table shows the mean absolute error (MAE) between the HR of the original and the reconstructed PPGs. MAE is calculated exclusively in the segment that corresponds to noise (2–120 s). Overall, the error remains below 2 bmp, with slight variations across the length of the noise. There is a tendency for lower error in longer segments, as minor variations in peak locations are better compensated there. On the other hand, when a reconstructed segment is longer, slight HR deviations are accumulated, which might lead to overall lower performance. Due to this opposed effects, there is no general rule on whether HR calculation is more efficient on shorter or longer reconstructed segments, while the error values are very close regardless of the length of noise.

Pearson correlation results between the original HR values and the HR values of the reconstructed PPGs for each noise length are illustrated in Figure 12. All values were statistically significant. As expected, correlations are very high in all cases. The HR exclusively in the reconstructed segment (HR-noise) shows higher values with increasing length of noise. This observation can be attributed to the higher effect of one extra pulse in a very small segment (such as 2 or 5 s) with respect to the effect of one extra pulse in a very long segment. The maximum difference between the number of pulses from the original and the reconstructed segments were 1 pulse for segments up to 45 s and 2 pulses for longer segments. The extra pulse or distorted HR was most often observed in the middle of the reconstructed segments.

BA plots for the HR calculated only on the noisy segment and for the SD of the HR can be seen in Figure 13 and Figure 14. Due to lack of space, 2 s long noisy PPG analysis has been omitted from the graphical data. The percentage of recordings that were found within the CI for each case and length can be seen in Table 5. Most features surpassed the 0.9 threshold for most of the lengths. The worst value was 0.86, seen only once in the 5 s segments of noise. Interestingly, longer segments of noise tend to show more trustworthy results, as indicated by the slightly higher percentages of recordings falling within the CI. This fact can be explained from the tendency of longer recordings to show a more similar mean HR and hence interval between successive peaks with the original PPG due to averaging. Besides the mean difference, the limits of CI are dependent on the SD of differences. Therefore, a high number of recordings falling within the CI suggests that the technique has a low SD of differences.

## 4. Discussion

MAs are probably the most common issue in PPG analysis [1]. With PPG sensors being extremely sensitive to even the slightest movement, and the MA spectrum overlapping with the HR spectrum, MA detection and removal has been a field of research in recent years, unfortunately with no satisfactory results [7,12,16,17,18,19,21]. This fact significantly complicates the task of maintaining consistently clean PPG signals, especially concerning endeavors such as estimating blood pressure from PPG data, which is a popular area of investigation [35]. The accuracy and fidelity of deriving ABP waveforms from PPG signals heavily rely on the absence of artifacts, particularly motion-induced distortions. Robust methods for identifying and eliminating motion artifacts are imperative to ensure the reliability and precision of the ABP waveform. Consequently, the continuous endeavor to attain and maintain clean PPG signals remains paramount in enhancing the efficacy and accuracy of physiological parameter estimation, such as ABP, through PPG-based models and analyses.

The vast majority of the algorithms elaborated in order to detect MAs are based on additional signals as a reference either for the corrupted PPG segment or in order to define the spectrum of the clean PPG signal and hence to differentiate it from the corrupted PPG segment [12,16,17,18,19,21,22,23]. Even in the case of MA detection without the use of additional signals, algorithms are limited either from the noise type that they can reliably detect or from the MAs’ usability. A recent study has been able to detect MAs without the use of additional signals or complex methodology [7]. Despite the fact that performance was very high for aperiodic signals, the use is limited to only detecting the artifacts, and there is no information of how the corrupted signal can be utilized. This is true for most of the developed techniques focused on the PPG artifact problem, where in the best case scenario they decide on the usability of the corrupted PPG but do not provide a solution on how to take advantage of it [12,16,17,18,22]. Some methods suggest post-processing the features under analysis in order to improve their calculation. This is performed by taking advantage of the information regarding the detected corrupted signal often using an additional signal as a reference [12,16,19]. In these cases as well, the corrupted PPG signal remains unused.

Signal reconstruction is another approach to the MA detection and removal problem. It is based on the principle of discarding the noisy components of the signal and keeping only the part related to biological activity, often using clean parts of the same signal as a reference. Such an approach has been suggested in the past, using singular spectrum analysis (SSA) [36]. Despite the high performance of this technique, significant limitations exist. Evaluation of the reconstruction was narrowed down to comparisons of the HR of original and reconstructed signals with a null-hypothesis test, neglecting the use of more adequate metrics such as error-based or BA analysis, while the algorithm can only be performed when the corrupted signal is partly usable. This limits the further utilization of long recordings due to some seconds of an almost completely lost signal. The most significant drawback of this method, however, is the computational complexity imposed by the multiple iterations needed to perform.

The present work suggests an alternative solution to MA confrontation, by introducing a simple technique able to restore the corrupted PPG segments by utilizing the information available on the clean PPG segments close to the noisy parts. The first part of this technique consists of an artifact detector, which can operate without the use of any additional recordings, a significant detail that expands the applicability of the algorithm. The algorithm performs spectral analysis in order to detect segments with a DF that exceeds the limits of the normal heartbeat or that do not fulfill specific conditions based on the normal behavior of the HR spectrum. It has been tested on real and synthetic noise, performing optimally in both cases.

The PPG signal reconstruction is performed under the hypothesis that the lost noisy part will share the same morphological properties with the clean signal that is adjacent to it and an HR that cannot exceed specific limits with respect to the HR observed some seconds before or after the artifacts. This assumption was based on the nature of the PPG signals, which present high stability and regularity across cardiac cycles. The additional consideration of the HRV of the adjacent clean segments surrounding the noisy part further allows for an HR adjustment of the reconstructed part, not only yielding high performance regarding HR estimation but also higher precision regarding morphological resemblance.

The performance of the algorithm has been verified on real noisy signals of a mean length of 13 s by comparing HR with the corresponding ECG segment and on two datasets of synthetic noisy signals by comparing the HR of the reconstructed PPGs with the real, clean PPGs. Regarding the real noisy signals, the HR in reconstructed PPGs showed a correlation higher than 99% with respect to the HR observed in the ECGs and the HR calculation with minimal differences, as confirmed by the error and BA analysis. Performance on synthetic PPGs was investigated by comparing the HR in 8 min PPGs with variable noise lengths, spanning from 2 s to 2 min. The rationale behind using variable noisy segments was, on one hand, to verify the applicability of the algorithm on various noise lengths and, on the other hand, to define an optimal duration threshold below which the reconstruction shows satisfactory results.

Overall satisfactory results have been observed for most of the features and noise lengths. Regarding the HR, median values of the entire 8 min recordings were almost identical for all cases. The error in the number of reconstructed pulses was at most 1 pulse for segments up to 45 s and 2 pulses for longer segments. At the same time, the extra pulse was always found in the middle of the reconstructed segment, close to the junction of the two reconstructed parts, corroborating the hypothesis that the possibility of erroneous PPG reconstruction augments with a higher distance from the clean segments. In any case, the difference in HR calculation for the reconstructed segments ($HR_{n}$) did not exceed 1.82 bpm. Correlation analysis further confirms these findings.

The results of BA analysis also indicated negligible differences in calculation between the original and the reconstructed PPGs for most of the segments, with the results falling within the set threshold of accepted values most of the time and the results that were outside those thresholds being close to them (1–4% difference). The aforementioned results pinpoint the high performance of the algorithm, able to reliably calculate HR values that are very close to the ground truth and with minimal errors, even in cases of long, noisy segments. Therefore, the presented algorithm can be an interesting alternative to the problem of MAs, allowing for continuous HR monitoring. The optimal length of noise up to which it can perform satisfactorily can be defined upon the needs of the studies and the device into which it can be implemented.

Although the proposed algorithm presents promising outcomes under specific conditions, it is crucial to acknowledge its limitations, particularly regarding its applicability in other scenarios marked by rapid or significant changes in the signal pattern. As already discussed, the key concept of the suggested methodology is the assumption of PPGs showing stable and repetitive patterns, allowing for the reconstruction of noisy segments by leveraging similarity with adjacent clean segments. The algorithm’s reliance on consistent patterns restricts its utility in other scenarios with rapidly changing signals, thereby limiting its effectiveness in capturing and reconstructing such dynamic fluctuations. Even though PPG signals, when not corrupted, are mostly characterized by repetitiveness, it could be possible that the presented technique lacks performance in extremely critical scenarios with very frequent abrupt changes in signal morphology or temporality, such as monitoring during intense exercise with very frequent movement artifacts or when attempting the reconstruction of other signal sources, characterized by rapid or significant changes. Due to the aforementioned reasons, the application of the proposed methodology should be performed with caution in order to ensure smooth and trustworthy results.

## 5. Conclusions

This work presents an alternative solution to MA detection and correction, by the reconstruction of the corrupted PPG signal based on simple principles. Its implementation is very straightforward and allows for uninterrupted health monitoring, facilitating the calculation of pivotal health markers. From wearable devices to investigation laboratories, the algorithm can be easily applied, significantly facilitating health tracking.

## Figures and Tables

**Figure 1 sensors-24-00141-f001:**
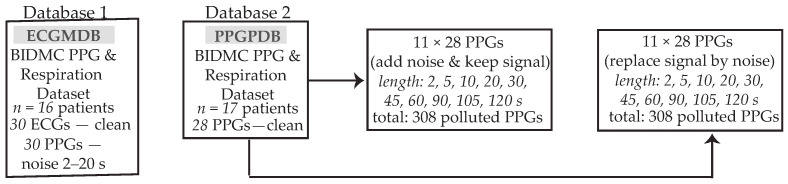
The two databases used. From the PPGPDB database, two different datasets were created, each containing 308 signals. The first dataset included synthetic signals created by adding noise of variable length to the originally clean PPG, while the second dataset included signals created by replacing segments of the originally clean PPG with noise of variable length, corresponding to the case where the signal is totally lost. ECG: electrocardiography; PPG: photoplethysmography.

**Figure 2 sensors-24-00141-f002:**
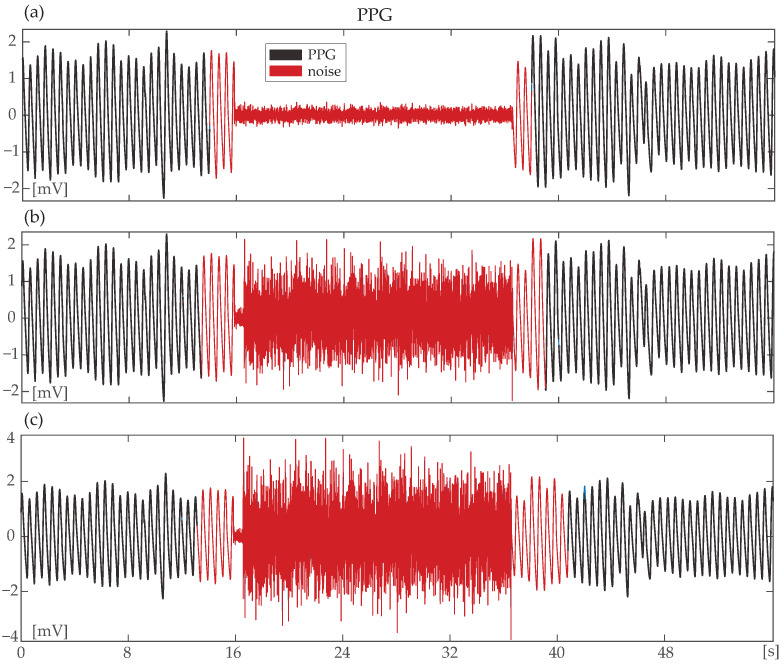
Motion artifact (MA) detection (red) in a synthetic signal with original signal cancellation. Noise is present from 16 to 36 s. (**a**) Amplitude of added noise is notably smaller than that of the clean segment; (**b**) Amplitude of added noise is close to that of the clean signal; (**c**) Amplitude of added noise is higher than the amplitude of the clean photoplethysmography (PPG) signal. In all three cases, the noise has been correctly identified.

**Figure 3 sensors-24-00141-f003:**
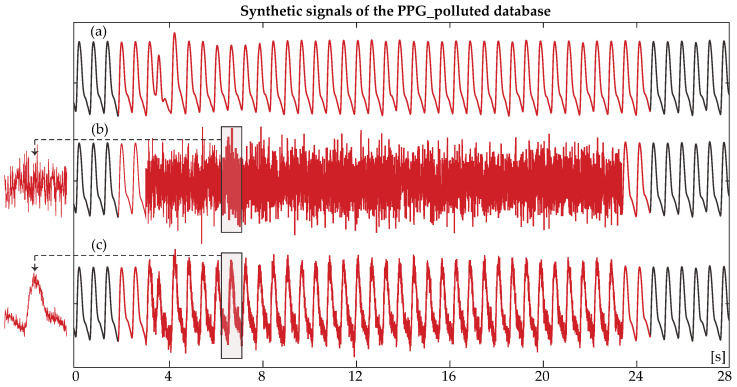
Synthetic signals derived from the originally clean PPGPDB. (**a**) An originally clean photoplethysmography (PPG) signal. Red signal indicates the segment that underwent noise replacement/addition; (**b**) The corresponding synthetic signal, where the clean segment was totally cancelled and replaced by noise (red); (**c**) The corresponding synthetic signal, where noise was added to the clean segment (red). In all cases, a second of clean signal was left on the left and right of the noise for the reconstruction process.

**Figure 6 sensors-24-00141-f006:**
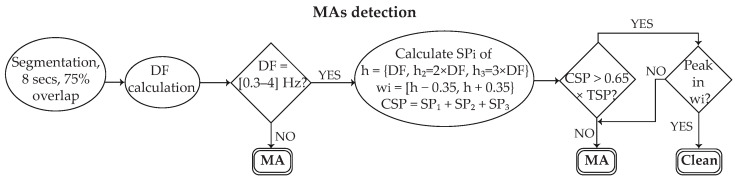
Block diagram of the main motion artifact (MA) detection algorithm before amplitude thresholds. Each 8 s segment passes through a three-level control (triangles) in order to be classified either as MA or as clean. DF: dominant frequency; SP: spectral power; CSP: cumulative spectral power.

**Figure 7 sensors-24-00141-f007:**
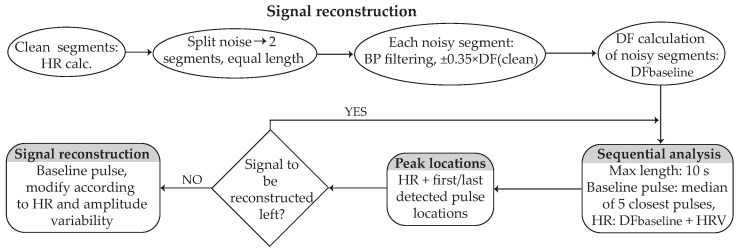
Block diagram of signal reconstruction. The first step is HR calculation in clean segments and DF in the noisy segment, after narrow band-pass filtering. An iterative process starts, defining the peak locations of the reconstructed segment from the DF of the noisy segment and the corresponding HRV of the closest clean/reconstructed segment. The maximum segment length of each iteration is 10 s. The final reconstruction is performed after peak locations have been defined, by the baseline pulse and the amplitude variability. HR: heart-rate; BP: band-pass; DF: dominant frequency; HRV: HR variability.

**Figure 8 sensors-24-00141-f008:**
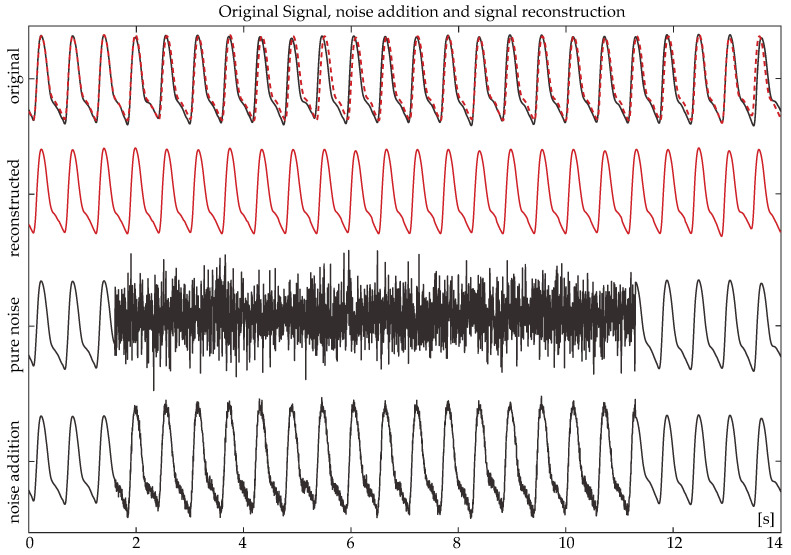
An originally clean photoplethysmography (PPG) signal (continuous line), the same PPG after the addition of 10 s of noise with (pure noise) and without (noise addition) original signal cancellation and the corresponding PPG segment after reconstruction (reconstructed—red). The red, dashed line in the original PPG is a projection of the reconstructed PPG, for comparison purposes.

**Figure 9 sensors-24-00141-f009:**
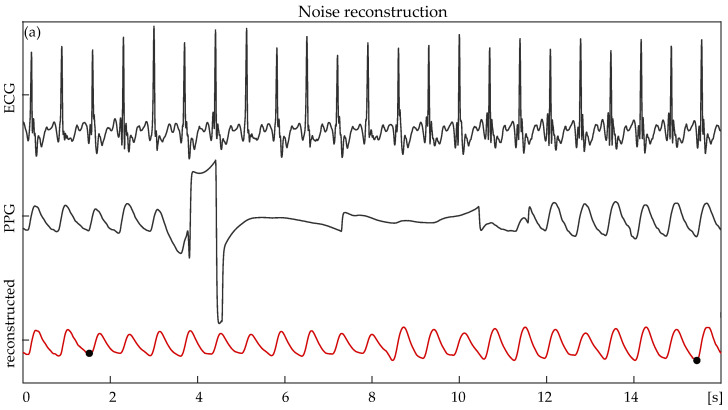
(**a**) A real, noisy photoplethysmography (PPG) signal (middle), the corresponding electrocardiography (ECG) segment (top), and the reconstructed PPG (bottom). The black dots in the reconstructed signal indicate the beginning and end points of the signal reconstruction. (**b**) Waveforms of (top) reconstructed (dashed, red) and ECG (continuous, black) signals and of (bottom) reconstructed (dashed, red) and noisy PPG signals (continuous, black) plotted together. The R peak–PPG peak ratio is maintained to a high degree in the reconstructed signal.

**Figure 10 sensors-24-00141-f010:**
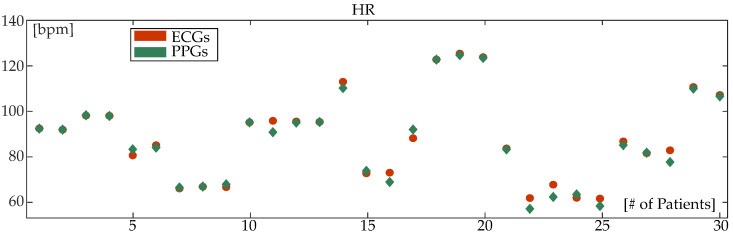
Heart-rate (HR) calculation from electrocardiography (ECGs) (red, circle) and the corresponding photoplethysmography (PPGs) (green, diamond) for the ECGMDB. HR is calculated over a mean duration of 13 s, with the minimum segment being 11 s and the maximum segment being 20 s.

**Figure 11 sensors-24-00141-f011:**
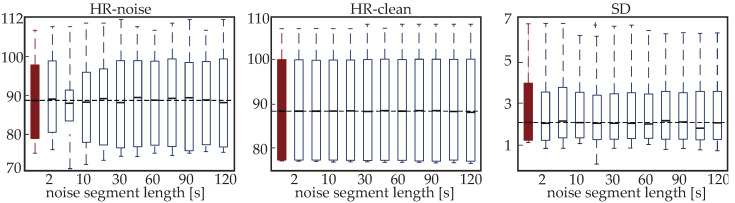
Box and whisker plots for the heart rate (HR) and standard deviation (SD) calculation of the original signal (dark red) and the reconstructed photoplethysmography (PPG) signals. Dashed line indicates the median value of the reference signal (original PPG-red boxplot). Features are measured in beats per minute (bpm).

**Figure 12 sensors-24-00141-f012:**
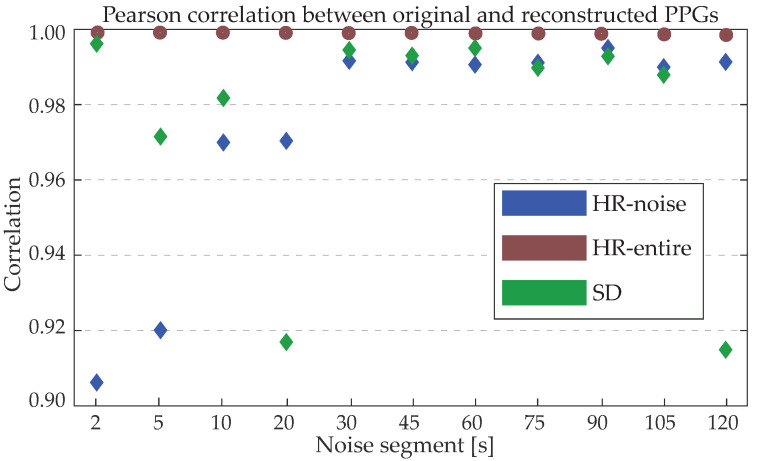
Pearson correlation [0–1] between the original and noise-corrupted photoplethysmography (PPGs) for the PPGPDB for all lengths of noise. All correlations were statistically significant (p<0.01).

**Figure 13 sensors-24-00141-f013:**
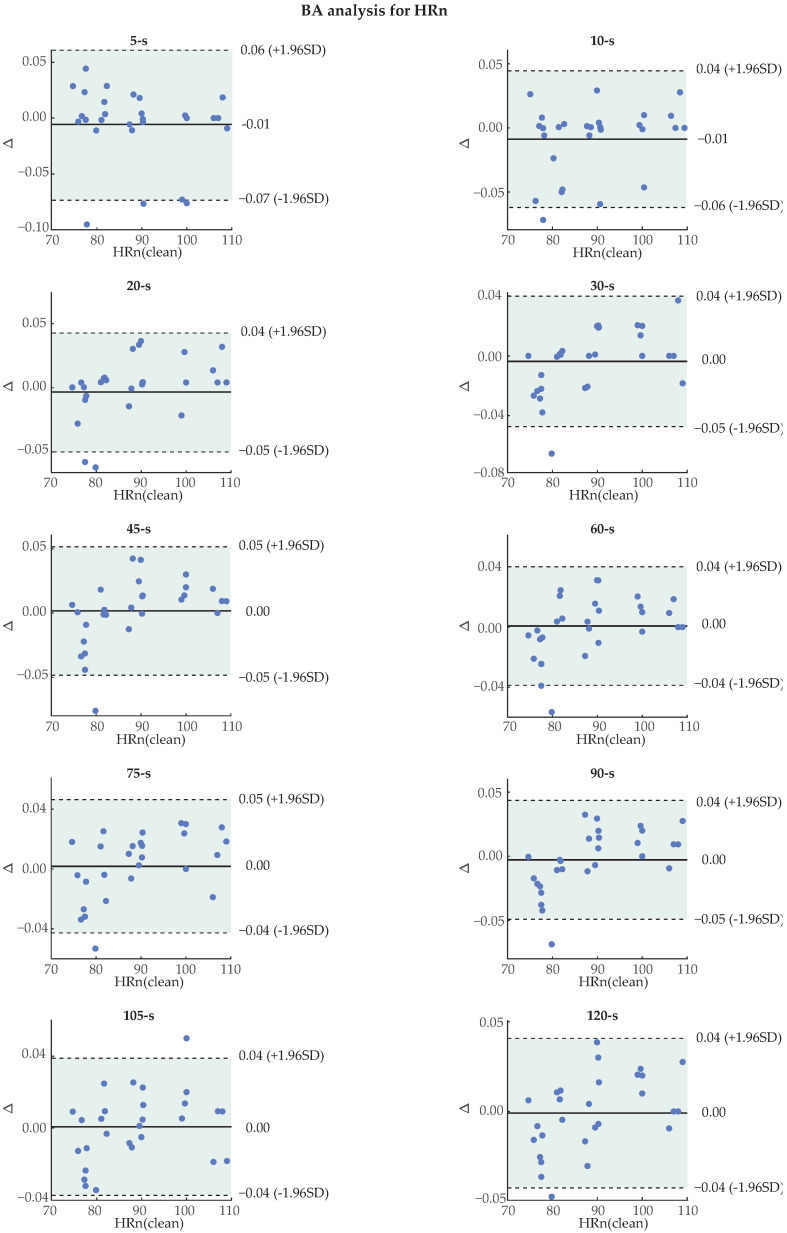
Bland–Altman (BA) plots for the heart rate (HR) of the segment corresponding to added noise, for all noise lengths. For each individual plot, Δ on the y-axis indicates the difference in bpm between the values of the original (x-axis) and the reconstructed photoplethysmography (PPG). The green, shaded area indicates the confidence interval (±1.96×SD). SD: standard deviation.

**Figure 14 sensors-24-00141-f014:**
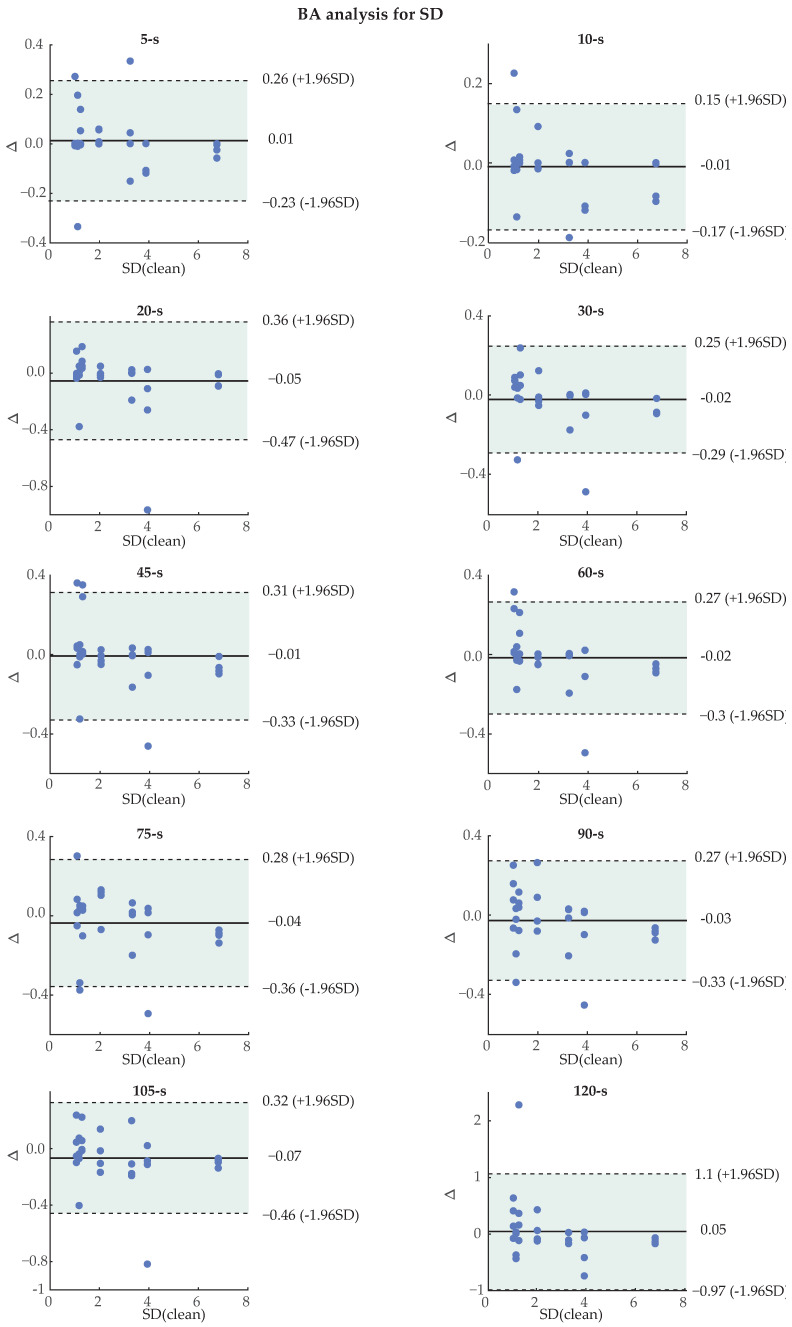
Bland–Altman (BA) plots for the standard deviation (SD) of the heart rate (HR), for all noise lengths. For each individual plot, Δ on the y-axis indicates the difference in bpm between the values of the original (x-axis) and the reconstructed photoplethysmography (PPG) signal. The green, shaded area indicates the confidence interval.

**Table 1 sensors-24-00141-t001:** Evaluation metrics for the ECGMDB and the subdatabase of the PPGPDB, where clean segments were completely cancelled and replaced by noise.

	Accuracy [%]	Sensitivity [%]	Specificity [%]
**ECGMDB**	90.91	93.75	88.24
**PPGPDB**	100.00	100.00	100.00

**Table 2 sensors-24-00141-t002:** Evaluation metrics for the PPGPDB subdatabase including signals with noise addition.

Noise Addition	Accuracy [%]	Sensitivity [%]	Specificity [%]
*2 s*	99.79	100.00	99.74
*5 s*	99.38	100.00	99.21
*10 s*	99.58	100.00	99.47
*20 s*	99.90	100.00	99.87
*30 s*	99.90	100.00	99.87
*45 s*	99.58	100.00	99.47
*60*	99.79	100.00	99.74
*75 s*	99.53	99.74	99.47
*90 s*	99.47	100.00	99.34
*105 s*	99.38	100.00	99.21
*120 s*	99.58	100.00	99.47

**Table 3 sensors-24-00141-t003:** Statistical results for HR values of the ECGMDB. For BA analysis, CI shows the percentage of HR values that are within the confidence interval. The HR of the ECGs are used as a reference. HR: heart rate; MWU: Mann–Whitney U-test; BA: Bland–Altman analysis; std: standard deviation; iqr: interquartile range; CI: confidence interval; CV: coefficient of variation; AE: absolute error.

	HR		Correlation		MWU	BA	
	Mean (std)	Median (iqr)	ρ [%]	p Value (ρ)	p Value	CI [%]	CV [%]
ECG	88.44(18.85)	87.50(25.29)	–	–	–	–	–
PPG	87.60(19.23)	88.01(29.04)	–	–	–	–	–
AE	1.59	0.671	–	–	–	–	–
ECG–PPG	–	–	99.31	<0.0001	0.8480	93.33	0.03

**Table 4 sensors-24-00141-t004:** Median values of the heart rate (HR) and the standard deviation (SD) of HR for the original (ref.) and the reconstructed photoplethysmography (PPGs) signals of variable length. $HR_{n}$ expresses the HR calculated only on the noisy segment, while $HR_{e}$ is the HR calculated over the entire 8 min signal. All measurements are in bpm. The mean absolute error (MAE) is between the noisy segment of the original (clean) and the reconstructed segment. The errors are calculated by only considering the noisy segment and not the entire recording.

	Median Values											
	**ref.**	**2 s**	**5 s**	**10 s**	**20 s**	**30 s**	**45 s**	**60 s**	**75 s**	**90 s**	**105 s**	**120 s**
$HR_{n}$	87.98	88.51	86.82	87.35	88.74	87.07	89.17	88.10	88.82	89.12	88.18	87.17
$HR_{e}$	88.10	88.10	88.10	88.15	88.16	88.04	88.23	88.10	88.23	88.23	88.03	87.84
SD	2.034	2.026	2.100	2.022	1.990	1.993	1.997	1.961	2.118	2.059	1.756	2.012
MAE	–	1.528	1.816	1.514	1.261	1.416	1.584	1.288	1.672	1.583	1.380	1.492

**Table 5 sensors-24-00141-t005:** Results of BA analysis for all noise lengths and all tested features. CI: percentage of recordings (0–1) within the confidence interval (CI). In most cases, almost 90% or above of recordings are found within CI.

	CI										
	**2 s**	**5 s**	**10 s**	**20 s**	**30 s**	**45 s**	**60 s**	**75 s**	**90 s**	**105 s**	**120 s**
$HR_{n}$	0.97	0.86	0.97	0.93	0.97	0.97	0.93	0.97	0.97	0.97	0.97
$HR_{e}$	0.93	0.93	0.93	0.93	0.97	0.89	0.97	0.93	0.93	0.93	1.00
SD	0.93	0.89	0.93	0.97	0.93	0.89	0.93	0.89	0.93	0.97	0.97

## Data Availability

The data supporting reported results and presented in this study are available from the corresponding author on request.

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
