# Peer review of "A Novel Signal Restoration Method of Noisy Photoplethysmograms for Uninterrupted Health Monitoring"

_sensors, 2023, doi:10.3390/s24010141_

Round 1

Reviewer 1 Report

Comments and Suggestions for Authors

General Comment:

The manuscript presents a novel approach to photoplethysmography (PPG) signal detection and reconstruction. Generally, the paper is well-written and technically sound. However, I have identified certain limitations in the method's assumptions and its performance benchmarking. Addressing the following concerns is crucial for a more comprehensive assessment of the algorithm's efficacy and applicability.

1.       My first concern is that the method heavily relies on the assumption that noisy segments closely resemble adjacent clean segments, as implied in the signal reconstruction steps (Section 2.3) and stated in Discussion section. This limits the application of the algorithm in scenarios with rapid or significant physiological changes.  Those dynamic signals to me are often more critical in health monitoring. I recommend implementing the algorithm on signals characterized by rapid or significant physiological changes. Subsequently, a detailed discussion on the algorithm's performance and limitations in these scenarios should be provided.

2.       The paper discusses various artifact detection/removal methods in the Introduction and Discussion sections. However, the manuscript lacks a direct comparison with these state-of-art methods using the same dataset. It is essential to include performance benchmarking against existing methods. This benchmarking should compare key metrics such as model accuracy, computational complexity, and real-time applicability.

Reviewer 2 Report

Comments and Suggestions for Authors

The article proposes an alternative approach for detecting and correcting motion artifacts (MAs) in photoplethysmography (PPG) signals. The proposed method involves reconstructing the corrupted PPG signal using simple principles, which makes it easy to implement and allows uninterrupted health monitoring. This technique also facilitates the calculation of pivotal health markers. The algorithm can be applied in various settings, from wearable devices to investigation laboratories, making health tracking significantly more accessible.

I highly recommend accepting this paper on PPG signal analysis for several compelling reasons. The authors have conducted a thorough and well-executed study that significantly contributes to the existing body of knowledge in the field.

Firstly, the paper exhibits a comprehensive literature review, demonstrating a strong understanding of the current state-of-the-art in PPG signal analysis. This foundational knowledge ensures the paper's relevance and situates it within the broader context of existing research. I would suggest the following article to the authors since it suggests a robust methodology that overcomes the limits due to occlusion and movements https://doi.org/10.1007/s12652-021-03635-6 . Here there is its improvement for mobile devices https://doi.org/10.1109/SMC53654.2022.9945406  I think they are strictly related to the topic addressed in their paper.

Secondly, the methodology employed in this study is robust and meticulously described. The authors have provided clear details on data collection, processing, and analysis techniques, allowing for reproducibility and validation of their findings. The transparency in the methodology enhances the credibility of the results.

I would suggest adding images (if present in data), showing the type of occlusion. Do the authors think that their results are general enough, or do they depend on the type of occlusion we could have in the original data?

Moreover, the results and findings presented in the paper are statistically sound and bring valuable insights into PPG signal characteristics and their potential applications. The authors have effectively discussed the implications of their results, paving the way for future research directions and practical applications in areas such as healthcare, biometrics, and human-computer interaction.

The paper also excels in its clarity of presentation and organization. The writing is concise, and the figures and tables are appropriately utilized to enhance the reader's understanding. This makes the paper accessible to a broad audience, including researchers, practitioners, and academics with varying expertise in PPG signal analysis.

In summary, the rigorous methodology, insightful findings, and clear presentation make this paper a strong candidate for acceptance. I believe it will make a valuable contribution to the scientific community and advance the understanding of PPG signal analysis, with potential applications in various domains.

Reviewer 3 Report

Comments and Suggestions for Authors

Overall, a good manuscript.

There are many flaws in the writing, which hinder the presentation of the article:

1. Abbreviation. Abbreviations in tables and figures must also be explained explicitly.

2. Non-parametric terms in formulas should not be italicized, such as RMS. \text can be used.

3. Why do some terms in Figure 6 begin with lowercase letters and others with uppercase letters (e.g., Peak)? Authors should take scientific writing seriously.

4. m should not be used as the unit of minutes. It has become the SI unit of length by convention.

5 Please note the following very relevant SOTA: https://doi.org/10.1016/j.bspc.2023.104972; Feature-Based Information Retrieval of Multimodal Biosignals with a Self-Similarity Matrix.

Moreover, the time series subsequence search library, an open-source code library used in many ECG and PPG research works, can also analyze or evaluate the task studied in this manuscript.

Comments on the Quality of English Language

See above.

Round 2

Reviewer 1 Report

Comments and Suggestions for Authors

The authors have provided reasonable responses to both my concerns. Now that the authors have agreed with the limitation that I mentioned,  I would like the authors to add a discussion to clearly state the limited scope of the the proposed algorithm. I would then recommend an acceptance of this paper. 

Reviewer 3 Report

Comments and Suggestions for Authors

I argue to accept the current version.
